# Fighting for Routes: Resource Allocation among Competing Planners in Transportation Networks

**Charlotte Roman** [1,*] **and Paolo Turrini** [2]

1  Mathematics Institute, University of Warwick, Coventry CV4 7AL, UK
2  Department of Computer Science, University of Warwick, Coventry CV4 7AL, UK; p.turrini@warwick.ac.uk
*  Correspondence: c.d.roman@warwick.ac.uk

**Abstract:** In transportation networks, incomplete information is ubiquitous, and users often delegate their route choice to distributed route planners. To model and study these systems, we introduce network control games, consisting of multiple actors seeking to optimise the social welfare of their assigned subpopulations through resource allocation in an underlying nonatomic congestion game. We first analyse the inefficiency of the routing equilibria by calculating the Price of Anarchy for polynomial cost functions, and then, using an Asynchronous Advantage Actor–Critic algorithm implementation, we show that reinforcement learning agents are vulnerable to choosing suboptimal routing as predicted by the theory. Finally, we extend the analysis to allow vehicles to choose their route planner and study the associated equilibria. Our results can be applied to mitigate inefficiency issues arising in large transport networks with route controlled autonomous vehicles.

**Keywords:** resource allocation; congestion games; multi-agent learning; efficiency





## 1. Introduction

Tackling traffic congestion has been a goal of many cities for decades, to, for example, reduce travel times and decrease air pollution. The increased adoption of automatic route planners such as GPS navigation, Google Maps, Waze, etc., can potentially have a positive impact on reducing congestion by the resource allocation of routes. A recent study demonstrated, to a large extent, that intelligent transport systems have a greater effect on improving congestion than building new roads [1]. Furthermore, the capacity for routing systems to control the flow of congestion is only increasing, and autonomous vehicle development will allow for routing control with minimal input from drivers.

Although arguably beneficial in many ways, the presence of multiple planners has important repercussions on the overall system efficiency, and the possibility for navigation applications to enforce socially desirable outcomes in transportation networks was recently listed as an open problem in Cooperative Artificial Intelligence [2].

Due to the size and complexity of the problem, using multi-agent reinforcement learning (MARL) algorithms for network control in order to optimise congestion could be a solution, but the problem remains of how far away the learnt strategies would be from the most socially desirable outcome. Before deploying such algorithms, we need to understand the *quality* of the resulting equilibria.

In Distributed Artificial Intelligence (DAI), congestion games [3] have become a reference model to analyse the inefficiency of traffic flows, with important implications for the design of better road systems [4]. In congestion games, self-interested players travel between origin and destination nodes in a network, choosing paths that minimise their travel time. Players' route choices constitute a Nash (or user) equilibrium when there is no incentive to unilaterally deviate, and then, they are typically compared against the total travel times, yielding the players' social welfare. The most often used measure of inefficiency is the Price of Anarchy (PoA) [5], which compares the worst Nash equilibrium routing with that of the optimal flow.

While Nash equilibria are important predictors, it is also well known that their assumptions on individuals' rationality are frequently not met in practice. In large transportation networks, it is often the case that individuals have incomplete knowledge of the network (see, e.g., the bounded rationality approaches in [6,7]) and rely on personal route planners to figure out their optimal route. An account of this intermediate perspective, where competing controllers act on the same selfish routing network, is still missing from the multi-agent systems literature.

### 1.1. Our Contribution

In this paper, we study intelligent routing systems that act as distributed controllers on a traffic network and analyse their impact on the overall efficiency. We develop a two-level game, called the network control game, where route planners have control over the resource allocation of an underlying nonatomic (information-constrained) congestion game. Specifically, each route planner controls a finite predetermined fraction of the total traffic by choosing information available to vehicles with the goal of minimising the travel time incurred by that fraction only. We show that this can be seen as a distributed resource allocation problem with separable welfare functions, where the resource sets are edges on a network and the strategies of a player must correspond to their given origin and destination pair. Modelling players of the nonatomic congestion game as vehicles and route controllers or planners as optimisers for a subset of vehicles, we prove that network control games are potential games and therefore have an essentially unique equilibrium. Then, we study equilibrium efficiency, showing that the PoA is highest when the allocation of vehicles to route planners is (approximately) proportional. We also give PoA bounds over polynomial cost functions, depending on the polynomial degree and the number of controllers. Furthermore, we give an example of an MARL implementation to show that this PoA occurs in practice. Finally, we extend network control games to allow vehicles to choose their route planner, showing that the unique resulting equilibrium has the highest total cost.

### 1.2. Paper Outline

We begin by discussing our work in connection with the related literature in Section 2 and outline the necessary preliminaries and notation in Section 3. Then, Section 4 introduces the network control games and studies the existence of equilibria. Section 5 calculates the PoA over polynomial cost functions, with an MARL implementation shown in Section 6. Finally, we analyse the extension where vehicles choose their route planner in Section 7.

## 2. Related Literature

Our work connects to a number of research lines in algorithmic game theory, focusing on the quality of equilibria in congestion games and resource allocation, and the research in DAI studying planning and control with boundedly rational agents.

Congestion games are a class of games first proposed by Rosenthal [3], which were utilised in research for modelling the behaviours of network systems. These were initially studied in the transportation literature by Wardrop [8] who established the conditions for a system equilibrium to exist when all travellers have minimum and equal costs. Their applications have increased to include many other situations that can be modelled with selfish players routing flow in a network, for example, machine scheduling or communication networks [9], as well as physical systems such as bandwidth allocation [10] or electrical networks [11]. However, their main application for congestion games is transportation [12–14].

Games, where the utilities of all players can be described with a single function, are called potential games [15], and these are, in fact, equivalent to congestion games. A useful property of potential games is that they always admit a pure Nash equilibrium. Finding a pure Nash equilibrium in an exact potential game is a PLS-complete problem [16]. However, improvement paths [15] converge at equilibrium for all potential games. For nonatomic

congestion games, i.e., ones with continuous player sets, the maximisers of the potential function are Nash equilibria of the potential game [17].

The PoA [5] was proposed as a measure of inefficiency representing the cost ratio of the worst possible Nash equilibrium to the social optimum. The PoA in network congestion games is a phenomenon that is independent of network topology [18]. The biased PoA [7] compares the cost of the worst equilibrium to the social optimum, when players have "wrong" cost functions, i.e., differing from the true cost due to biases or heterogeneous preferences.

From the point of view of distributed control, an important related model is Stackelberg routing games, where a portion of the total flow is controlled centrally by a "leader", while the "followers" play as selfish vehicles. Stackelberg routing was first proposed by [19], characterising which instances are optimal. Roughgarden [20] found the ratio between worst-case and best-case costs in these games, and the impact of Stackelberg routing on the PoA has also been established for general networks [21]. Single-leader Stackelberg equilibria in congestion games have been looked at, and it is known that they cannot be approximated in polynomial time [22]. Instead, multi-leader Stackelberg games are largely unexplored in this context [23]. Our approach features multiple leaders but not Stackelberg-like "followers", which impacts our results on the PoA.

Much of the transport literature is aimed at reducing congestion, and increasing efficiency in traffic networks focuses on introducing tolls [24–26]. Information design, which is closely related to our approach, has more recently been considered as a mechanism to reduce congestion [6,7,27]. The information-constrained variant of a nonatomic congestion game was first introduced to show that information could cause vehicles to change their departure times in such a way as to exacerbate congestion rather than ease it [28]. The set of outcomes that can arise in equilibrium for some information structure is equal to the set of Bayes correlated equilibria [29]. Das et al. [30] considered an information designer seeking to maximise welfare and restore efficiency through signals using information design. Tavafoghi and Teneketzis [31] showed that the socially efficient routing outcome is achievable through public and private information mechanisms. Moreover, Ikegami et al. [32] consider a centralised mediator to recommend routing to users taking into account their preferences for incomplete information games. Our work differs from the private information design literature. In our model, route planners control the routing rather than provide signals, and multiple agents attempt to optimise 'group' welfare.

A similar game is the splittable congestion game, which was first studied in the context of communication networks [9]. Here, each player in the congestion game assigns a weight to the possible strategies that arise when considering coalitions of players in nonatomic congestion games. The bounds on the PoA for splittable congestion games have been found for polynomial cost functions [33], which have the same bound when there are an infinite number of route planners in a network control game.

Network control games can be seen as resource allocation games, where the resources are edges in a network and the potential function is the total cost of players' travel times. Distributed resource allocation problems aim to allocate resources for optimal utilisation, such as distributed welfare games [34] and cost-sharing protocols [35]. A recent survey of game-theoretic control of networked systems highlights major advancements and applications [4].

Additionally, distributed welfare games [34] utilise game-theoretic control for distributed resource allocation where the distribution rule is chosen to maximise the welfare of resource utilisation. Different distribution rules can be compared by their desirable properties such as scalability, the existence of Nash equilibria, PoA, and Price of Stability. In this context, protocols have been studied to improve the equilibria of network cost-sharing games [35], while [36] studied welfare-optimising designers under full and partial control. We consider a distributed resource allocation problem with separable welfare functions where the resource sets are edges on a network and the strategies of player must coincide with their given origin and destination pair, i.e., on a congestion game.

Regulating the flow of traffic in complex road networks is an important application of artificial intelligence technologies usually involving distributed optimisation and multi-agent learning methods [37–39]. Most of the literature focuses on adapting traffic lights to coordinate traffic, but MARL can also be used to improve traffic flow through resource allocation, as we show in Section 6.

## 3. Preliminaries

We begin by introducing some standard definitions of nonatomic congestion games and the properties of their equilibria.

Let $N = \{1, \ldots, n\}$ be a nonempty finite set of player (or vehicle) populations such that players in the same population share the same available route choices (or strategy set). For population $i \in N$, the *demand* for a population, i.e., the traffic volume associated with that population, is $d_i > 0$. Each population $i$ has a nonempty finite resource set $E_i$ made up of *relevant* resources, i.e., those edges that are used in at least one route choice, $S_i \subseteq 2^{E_i}$, where $S_i$ is the strategy set of $i$. Denote $E$ as the *irredundant* resource set $E = \bigcup_{i \in N} E_i$, the set of edges used in any strategy set. Resource cost functions, $c_e : \mathbb{R}_{\geq 0} \to \mathbb{R}_{\geq 0} \cup \{\infty\}$ such that $e \in E$, are assumed to be continuous, nondecreasing, and non-negative.

We assume that individuals have limited knowledge of the routing options; i.e., we assume there exist $K_i \geq 1$ information types in each population $i \in N$ and refer to a player from population $i$ of type $k$ as $(i, k)$, where the demand for a type is $d_{(i,k)} \geq 0$. Information types can restrict knowledge of the resources; i.e., each population–type pair is associated with a known set $E_{(i,k)} \subseteq E_i$. Formally, a *nonatomic information-constrained* (NIC) *congestion game* is defined as a tuple $(N, (K_i), (E_{(i,k)}), (S_{(i,k)}), (c_e)_{e \in E}, (d_{(i,k)}))$, with $i \in N, k \in K_i$.

The outcome of all players of type $k$ choosing strategies leads to a vector $\boldsymbol{x}^{(i,k)}$ satisfying

$$\sum_{s \in S_{(i,k)}} x_s^{(i,k)} = d_{(i,k)} \tag{1}$$

and $x_s^{(i,k)} \geq 0, \forall s \in S_{(i,k)}$. A strategy distribution or outcome $\boldsymbol{x} = (\boldsymbol{x}^{(i,k)})_{i \in N}$ is *feasible* if Equation (1) holds $\forall i \in N$ and $\forall k \in K_i$. Then, denote the load on $e$ in an outcome $\boldsymbol{x}$ to be $f_e(\boldsymbol{x}) = \sum_{i \in N} \sum_{s \in S_{(i,k)}} x_s^i \mathbb{1}_s(e)$, where $\mathbb{1}$ is the indicator function. In a strategy distribution, $\boldsymbol{x} := (\boldsymbol{x}^{(i,k)})_{k \in K_i, i \in N}$, a player of knowledge type $(i,k)$ incurs a *cost* of $C_{(i,k)}(s, \boldsymbol{x}) := \sum_{e \in s} c_e(f_e(\boldsymbol{x}))$ when selecting strategy $s \in S_{(i,k)}$. An *information-constrained user equilibrium* (ICUE) [6] is a strategy distribution $\boldsymbol{x}$ such that all players choose a strategy of minimum cost: $\forall i \in N, k \in K_i$ and strategies $s, s' \in S_{(i,k)}$ such that when $x_s^{(i,k)} > 0$ we have $C_{(i,k)}(s, \boldsymbol{x}) \leq C_{(i,k)}(s', \boldsymbol{x})$. Every player of the same knowledge type has the same cost at a UE $\boldsymbol{x}$, which is denoted $C_{(i,k)}(\boldsymbol{x})$. We say that a user equilibrium is *essentially unique* if all user equilibria have the same social cost. For any nonatomic congestion game, there exists a user equilibrium, and it is essentially unique [40].

The *social cost* of $\boldsymbol{x}$ is the total cost incurred by all players,

$$SC(\boldsymbol{x}) := \sum_{i \in N} \sum_{(k \in K_i} C_{(i,k)}(\boldsymbol{x}) d_{(i,k)} \tag{2}$$

Strategy distribution $\boldsymbol{x}$ is a *social optimum* (SO) if it minimises Equation (2). Formally, a SO solves $\min_{\boldsymbol{x}} SC(\boldsymbol{x})$, such that $\sum_{s \in S_{(i,k)}} x_{(i,k)}^s = d_{(i,k)}, \forall i \in N, k \in K_i, x_{(i,k)}^s \geq 0$.

In most cases, the SO solution is different to the UE solution, since players only maximise their individual utility. Pigou [41] was the first to show this on a network with one origin and one destination and two parallel edges joining them, for a population of size 1. The cost of the first of the edges is constant at 1, and the second costs the same as the proportion of players that choose it. The UE here is for all players to use the second edge, which gives a social cost of 1, whereas the optimal routing is to split players equally along edges for a social cost of $\frac{3}{4}$.

The efficiency of the UE when compared with the SO is the *Price of Anarchy* (PoA). It is defined as the ratio between the social cost of a SO outcome and the worst social cost of a UE,

$$PoA = \frac{\max_{y \in UE} SC(y)}{\min_x SC(x)} \tag{3}$$

where $UE$ is the set of user equilibria. For example, in Pigou's network, the PoA is $\frac{4}{3}$.

An *exact potential game* is one that can be expressed using a single global payoff function called the *potential function*. More formally, a game is an exact potential game if it has a potential function $\Phi : A \to \mathbb{R}$ such that $\forall a_{-i} \in A_{-i}$, $\forall a_i, a_i' \in A_i$, $\Phi(a_i, a_{-i}) - \Phi(a_i', a_{-i}) = u_i(a_i, a_{-i}) - u_i(a_i', a_{-i})$. Here, the notation $-i$ means all players in $N$ excluding $i$, i.e., $\{1, .., i-1, i+1, .., |N|\}$. The concept of potential games was first posed by [15] for atomic games and later extended to nonatomic games [17,42]. Potential games and congestion games are equivalent, where a player's utility is their negative cost. The potential function for nonatomic congestion games is

$$\Phi(x) := \sum_{e \in E} \int_0^{f_e(x)} c_e(z) dz \tag{4}$$

where $x$ is the strategy distribution of players, which is also referred to as the Beckmann function [43]. A strategy distribution is an ICUE if, and only if, it minimises the potential function [6] (an extension of results in [40,43]).

When studying the PoA in network control games, we will turn our attention to social dilemmas, i.e., games in which there exists a conflict between individual and social preferences. The classic two-player matrix game social dilemma is a game with payoffs shown in Table 1, where we must have the conditions $R > P$, $R > S$, $2R > T + S$, and either $T > R$ or $P > S$. A social dilemma, by definition, has a PoA strictly greater than 1.

**Table 1.** The classic format of social dilemma payoffs in a two-player matrix game, where $(A, B)$ means that player 1 receives payoff $A$ and player 2 receives payoff $B$.

|   | $C$ | $D$ |
|---|---|---|
| $C$ | $(R, R)$ | $(S, T)$ |
| $D$ | $(T, S)$ | $(P, P)$ |

Learning algorithms can be used to find strategies in large or complex environments. In this paper, we consider the effects of using reinforcement learning algorithms to solve the problem of route control in information-constrained nonatomic congestion games.

Reinforcement learning (RL) is a framework where an agent earns rewards for taking actions in a given environment. The goal is to find a policy—a sequence of actions to take in each environmental state—in order to maximise rewards. Value functions are used to estimate long-term rewards given that the agent observes a particular state and selects actions aligning with its policy. Equivalently, this can be formalised as a single-player stochastic game where the policy is the agent's strategy. See [44] for a detailed introduction to RL.

The environment is represented by a state variable, $s \in S$, and the principle task of the agent is to select the best action, $a \in A$, given the current state. An optimal policy states the actions to be taken in a given state to achieve the highest expected return.

A *Markov decision process* (MDP) is a discrete-time stochastic control process that provides a suitable mathematical framework for modelling an agent's reasoning and planning strategies in the face of uncertainty. It satisfies the Markov property—the probability distribution over the next set of states only depends on the current state and not its history. Formally, we write an MDP as a tuple $(S, A, P, R)$: state space $S$, action set $A$, Markovian transition model $P : S \times A \times S \mapsto [0, 1]$, and reward function $R : S \times A \times S \mapsto \mathbb{R}$.

The goal of the agent is to select a sequence of actions, or *policy* $\pi$. An *optimal policy* $\pi^*$ maximises the cumulative discounted return, $G_t = \sum_{k=0}^{\infty} \gamma^k r_{t+k+1}$, where $\gamma \in [0,1]$ is a discount factor and $r_i = R(s_i, a_i, s_{i+1})$ is the reward at step $i$. The state-value function, or *value function*, $V_\pi : S \to \mathbb{R}$ describes the expected value of following policy $\pi$ from state $s$, $V_\pi(s) := E_\pi[G_t|s_t = s]$.

The action–value function, or *Q-function*, $Q_\pi : S \times A \to \mathbb{R}$ estimates the expected value of choosing an action $a$ in state $s$ then following policy $\pi$: $Q_\pi(s,a) = E_\pi[G_t|s_t = s, a_t = a]$. We can write the Q-function in terms of the value function as follows:

$$Q_\pi(s,a) = R(s,a,s') + \gamma \sum_{s'} P(s,a,s') V_\pi(s') \tag{5}$$

In MARL, each agent must make assumptions about their opponents' strategies in order to optimise their own payoff. An MDP can be generalised to capture multiple agents through the use of Markov games. Formally, a *Markov game* is a tuple $(N, S, (A^i)_{i \in N}, P, (R^i)_{i \in N}, \gamma)$ where $N$ is the set of agents, state space $S$, $A^i$ is the action space of $i \in N$, $P : S \times A^1 \times ... \times A^{|N|} \times S \to [0,1]$ is the transition function, $R^i : S \times ... \times A^{|N|} \times S \to \mathbb{R}$ is the reward function, and $\gamma$ is the discount factor.

Denote the action profile of agents at time $t$ is $\boldsymbol{a}_t$, then we can define the value function for player $i$ as

$$V^i_{\pi^i, \pi^{-i}}(s) := E_{\pi^{-i}}\left[\sum_{t \geq 0} \gamma^t R^i(s_t, \boldsymbol{a}_t, s_{t+1}) | a^i_t \sim \pi^i(.|s_t), s_0 = s\right] \tag{6}$$

For Markov games, a *Nash equilibrium* is a joint policy $\boldsymbol{\pi} = (\pi^1, \ldots, \pi^{|N|})$ such that for all $i \in N$ and $s \in S$,

$$V^i_{\pi^i, \pi^{-i}}(s) \geq V^i_{\bar{\pi}^i, \pi^{-i}}(s) \text{ for all } \bar{\pi}^i.$$

Actor–critic methods [45] are a class of algorithms where a 'critic' advises an 'actor' of the quality of each action. The actor and critic each learn separately; the critic estimates the value function, while the actor learns the policy based on feedback from the critic.

Asynchronous Advantage Actor–Critic (A3C) is a model-free policy optimisation-based MARL algorithm. It equips the actor–critic format with independent local agents (asynchronicity) whereby the critics estimate the *advantage function*, which is defined as the Q-function Equation (5) minus the value function Equation (6). In policy optimisation, we learn the policy directly rather than the Q-values. In deep learning, we learn the parameters $\theta$ of the neural network that represents the policy or value function. For further details of the algorithm, see [46].

## 4. Network Control Games

Suppose that the routing choices of vehicles in an NIC congestion game $\mathcal{M}$ are controlled by a set of route planners $R$, where each route planner aims at minimising the total travel cost of the (nonempty) portion of vehicles assigned to them $N_r$, where $r \in R$ and $N_r \subseteq N$. For instance, this could occur in a setting of competing autonomous taxi companies within a city. Each taxi company wishes to minimise the journey times of their fleet for customer satisfaction and energy efficiency but does not use the same routing systems as the other companies.

The way in which the route planners have control over the routing choices is by choosing which knowledge set is available to each player, i.e., any type $k \in K_i$ for a vehicle in population $i$. Since there exist knowledge types that have only one route available as a strategy, this action space is a superset of controlling the routes of vehicles. Thus, we allow route planners to control the demand for each knowledge type within the fraction of flow they control; i.e., they control the distribution of knowledge types within populations. Choosing this action space allows for a more generalised model than if they selected a single route for each vehicle. For instance, a navigation app would give its users a choice between only a few routes; thus, drivers have incomplete knowledge of the network available

to them. For example, autonomous vehicles may not give their passengers a choice of route. In this case, the knowledge set would contain only the route that the autonomous vehicle follows.

Let the size of each population $i \in N$ controlled by $r \in R$ be denoted $d_i^r$, where $\sum_{r \in R} d_i^r = d_i$ and $d_i^r = 0$, for $i \notin N_r$. We can view the game as an information design problem, where a player $r$ partitions populations in $N_r$ into sets of information types to minimise the social cost of $N_r$. The route planner $r$ has a strategy set denoted $\boldsymbol{K}_r := (K_i)_{i \in N_r}$. Thus, a route planner chooses the information type demands $d_{(i,k)}$, such that $\forall i \in N_r$, $\sum_{(i,k) \in (\boldsymbol{K}_r)_i} d_{(i,k)} = d_i^r$. Let the strategy space for route planners be $D_{\kappa_r}$ where $\kappa_r$ is the set of all information sets $K_i$ for any $i \in N_r$. Moreover, for any $\boldsymbol{d} \in D_{\kappa_r}$ and $\forall i \in N_r$, we have $\sum_{(i,k) \in \kappa_r} d_{(i,k)} \mathbb{1}_{K_i}(i,k) = d_i^r$, where $\mathbb{1}$ is the indicator function. Let the combined strategy space of all route planners be denoted as $D_\kappa$, where $\kappa$ is the set of all information types.

Now, we define a *network control game* to be a tuple $(\mathcal{M}, R, (N_r)_{r \in R}, (d_i^r)_{i \in N_r}, (D_{\kappa_r})_{r \in R})$, where $\mathcal{M}$ is an NIC congestion game, $R$ is the set of route planners, $N_r$ is the population controlled by $r \in R$, $d_i^r$ is the demand of population $i$ controlled by $r$, and $D_{\kappa_r}$ is the strategy space of $r$.

Let the *share of control* of route planner $r$ be $\frac{\sum_{i \in N_r} d_i^r}{\sum_{i \in N} d_i}$. If a route planner has a share of control equal to 1, then we say it has *full control* of the game. The control of $r$ over a population $i$ is defined as $\frac{d_i^r}{d_i}$. If $\forall r \in R$ and $\forall i \in N$, the control of $r$ over population $i$ is $|R|^{-1}$, then we say that the game is *proportional*.

Observe now that the outcome of all route planners' strategies $\boldsymbol{d} := (\boldsymbol{d}^r)_{r \in R}$ leads to an ICUE $\boldsymbol{x}$ in the underlying game. This is true, since it is a two-level game, where route planners first choose the information types, and then, the congestion game is played by vehicles in the second stage. Given this, the cost function of a route planner $C_r$ : $D_{\kappa_r} \to \mathbb{R}_{\geq 0}$ is defined as $C_r(\boldsymbol{d}^r, \boldsymbol{d}^{-r}) := \sum_{(i,k) \in \kappa_r} C(i,k)(\boldsymbol{x}) d_{(i,k)}^r \ \forall r \in R$, where $\boldsymbol{x}$ is the ICUE from $(\boldsymbol{d}^r, \boldsymbol{d}^{-r})$. Note that due to the notation $-r$, we can use $C_r(\boldsymbol{d})$ and $C_r(\boldsymbol{d}^r, \boldsymbol{d}^{-r})$ interchangeably.

Then, an outcome $\boldsymbol{d}$ is a Nash equilibrium of the network control game if, and only if, $\forall r \in R$, we have $C_r(\boldsymbol{d}) \leq C_r(\boldsymbol{d}', \boldsymbol{d}^{-r}) \ \forall \boldsymbol{d}' \in D_{\kappa_r}$. We can show the existence of Nash equilibria in network control games by showing that these are exact potential games.

**Proposition 1.** *A network control game is an exact potential game for potential $\Phi$ defined as*

$$\Phi(\boldsymbol{d}) := \sum_{e \in E} \int_0^{f_e(\boldsymbol{x})} c_e(z) dz,$$

*where $\boldsymbol{x}$ is the ICUE formed from $\boldsymbol{d}$.*

**Proof.** Consider a unilateral deviation $\hat{\boldsymbol{d}}^r$ of route planner $r$ from an outcome $\boldsymbol{d}$ with respective ICUE profiles $\hat{\boldsymbol{x}}$ and $\boldsymbol{x}$.

$$\Phi(\hat{\boldsymbol{d}}^r, \boldsymbol{d}^{-r}) - \Phi(\boldsymbol{d}) = \sum_{e \in E} \int_0^{f_e(\hat{\boldsymbol{x}})} c_e(z) dz - \sum_{e \in E} \int_0^{f_e(\boldsymbol{x})} c_e(z) dz \tag{7}$$

Since the deviation from $\boldsymbol{x}$ to $\tilde{\boldsymbol{x}}$ only involves edges in $\kappa_r$, we can rewrite the right-hand side of Equation (7) as

$$= \sum_{(i,k) \in \kappa_r} \sum_{e \in s_{(i,k)}} \left[ \int_0^{f_e(\hat{\boldsymbol{x}})} c_e(z) dz - \int_0^{f_e(\boldsymbol{x})} c_e(z) dz \right]$$

$$= \sum_{(i,k) \in \kappa_r} \left[ C_{(i,k)}(\hat{\boldsymbol{x}}) - C_{(i,k)}(\boldsymbol{x}) \right]$$

$$= C_r(\hat{\boldsymbol{d}}^r, \boldsymbol{d}^{-r}) - C_r(\boldsymbol{d})$$

Thus, the function $\Phi$ is an exact potential function. By definition, the network control game is an exact potential game. $\square$

Since we have an exact potential game with nondecreasing edge costs, Corollary 1 follows directly from Acemoglu et al. [6] (Theorem 1).

**Corollary 1.** *For every network control game, there exists a Nash equilibrium, and it is essentially unique.*

As network control games are exact potential games, all of the results known for nonatomic congestion games will also hold in the new context, e.g., [18,47]. Nonetheless, these games allow for insights into how the distribution of vehicle route planners will affect traffic equilibria, which is an important and novel contribution to the literature (see Section 2 for an in-depth discussion).

We now define the PoA of a network control game as

$$PoA = \frac{\max_{d \in NE} C(d)}{\min_{d \in D_\kappa} C(d)},$$

where $NE$ is the set of Nash equilibria. Since there is a one-to-one mapping of flow to route planners, the social cost is defined the same as the underlying congestion game.

Note that our setup can be extended to incorporate vehicles that are not fully controlled by a route planner, e.g., by allowing route planners that give full information sets to their populations. However, we only consider vehicles following a route planner directly to more easily classify the best and worst-case equilibria from the full route control of populations.

We also note that for any strategy distribution in a (information-constrained) nonatomic congestion game, we can, without loss of generality, restrict ourselves to pure strategy equivalents. A route planner has no incentive to recommend multiple routes to a vehicle, since this creates uncertainty about vehicle route choice. Thus, henceforth, we study the case where all information sets chosen by the route planners contain only one strategy. As such, the profile set by the route planners $d$ has a deterministic associated ICUE $x$.

## 5. Inefficiency of Route Controllers

To see how the network control game creates inefficient equilibria, first consider what happens as we change the number of route planners in a proportional game. First, suppose that a route planner has full control of the game, then all vehicles follow the same route planner. Thus, the route planner has an objective function equal to the social cost of the system: $C_r(d) = \sum_{(i,k) \in \kappa_r} C_{(i,k)}(x) d_{(i,k)} = \sum_{r \in R} \sum_{(i,k) \in K_r} C_{(i,k)}(x) d_{(i,k)} = SC(x)$. As such, the case with $|R| = 1$ will implement the socially optimal routing allocation.

Now, as we increase the number of route planners, the demand of the population controlled by a single player decreases. As $|R| \to \infty$ and since the game is proportional, we have that $d_{N_r} \to 0, \forall r \in R$. As we now have an infinite number of agents controlling a negligible amount of flow, we are back to a simple NIC congestion game. This occurs since $C_{-r}(d^r, d^{-r}) = C_{-r}(d^{-r}) \, \forall d^r \in D_\kappa$, when the proportion of control of $r$ is negligible. The PoA of the game is now the same as in its underlying NIC congestion game. Thus, if there is more than one route planner in a proportional network control game, which is true by definition, there is an inefficient equilibrium if the NIC congestion game admits one.

**Proposition 2.** *A proportional network control game has a PoA strictly greater than 1 if, and only if, the underlying NIC congestion game does.*

**Proof.** ($\Rightarrow$). If the PoA of a network control game is strictly greater than 1, then there is an incentive to choose suboptimal routing at the Nash equilibrium. As the number of route planners in the proportional game tends to infinity, we approach the underlying NIC

congestion game. As such, the suboptimal routing exists in the NIC congestion game, and so, the PoA for the NIC congestion game is strictly greater than 1, too.

($\Leftarrow$) If the PoA of an NIC congestion game underlying a proportional network game is strictly greater than 1, then we know that at the UE, there exists a suboptimal selfish routing of drivers. Let the Nash equilibrium of routing be $D$ and the SO be $C$. Since the game is proportional, all route planners have the same strategy space. Thus, $\forall r \in R$, (i) $C_r(D) \leq C_r(C_r, D_{-r})$, (ii) $SC(C) \leq SC(D)$, and (iii) $SC(C) \leq SC(C_r, D_{-r})$.

We can write this as a two-player ($r$ and $-r$) normal form game with the actions $C_r$, $D_r$, $C_{-r}$, and $D_{-r}$. This, combined with the inequalities above, indicates that the payoffs comply with the conditions required for a social dilemma. As such, the PoA is strictly greater than 1. $\square$

Thus, we have identified that having multiple route planners controlling the flow on any NIC congestion game with an inefficient equilibrium will not have SO equilibrium flow. We will now bound this inefficiency by finding the PoA of the route control game.

Since the PoA is independent of network topology in all nonatomic congestion games [18], we can use the Pigou example to illustrate the inefficiency of having multiple route planners. We assume that cost functions are polynomial with maximum degree $p$. To begin, let us consider linear cost functions, i.e., $p = 1$.

**Example 1** (Two route planners). *Suppose we have a total flow of 1 and two route planners 1 and 2 with respective population control of $d^1$ and $d^2 = 1 - d^1$ on a Pigou network. Each route planner must solve the following minimisation problem to find their equilibrium routing defined by the variable $x_r$ for $r \in \{1, 2\}$ $r \neq s \in \{1, 2\}$, as defined in Figures 1 and 2.*

$$\min_{x_r} \left[ x_r(x_r + x_s) + (d^r - x_r) \right]$$

*subject to $0 \leq x_r \leq d^r$. This gives us the Lagrangian function (where $s \in \{1, 2\}$, $s \neq r$):*

$$L(x_r, \lambda_1, \lambda_2) = x_r(x_r + x_s) + d^r - x_r - \lambda(d^r - x_r)$$

*The corresponding Karush–Kuhn–Tucker conditions are:*

$$\frac{\delta L}{\delta x_r} = 2x_r + x_s - 1 + \lambda = 0 \quad \lambda(x_r - d^r) = 0$$

*Consider the following three cases:*

1. *First, consider the case where $x_r = d^r$. Since $\lambda \geq 0$, we must have $d^r \leq \frac{(1 - x_s)}{2}$. Route planner $r$ plays selfishly by routing along the bottom edge only if their control is small.*
2. *Now, suppose that $x_r \neq d^r$ and $x_s \neq d^s$. The solution here is $x_r = x_s = \frac{1}{3}$.*
3. *The last possible case is where $x_s = d^s$, and similarly, this occurs when $d^s \leq \frac{1}{2}(1 - x_r)$.*

*Thus, the optimal routing of splitting the vehicles equally between routes only occurs when there is one route planner with full control. The social cost of equilibria is shown in Figure 3.*

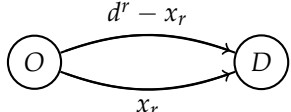

**Figure 1.** The strategy for route planner $r \in R$ where $x_r \in [0, d^r]$ on a Pigou network with two route planners, $r$ and $s$.

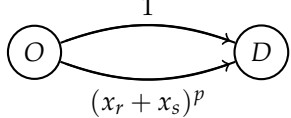

**Figure 2.** The edge cost functions, where $p \geq 0$ and $x_r$ and $x_s$ are defined from the flows in Figure 1.

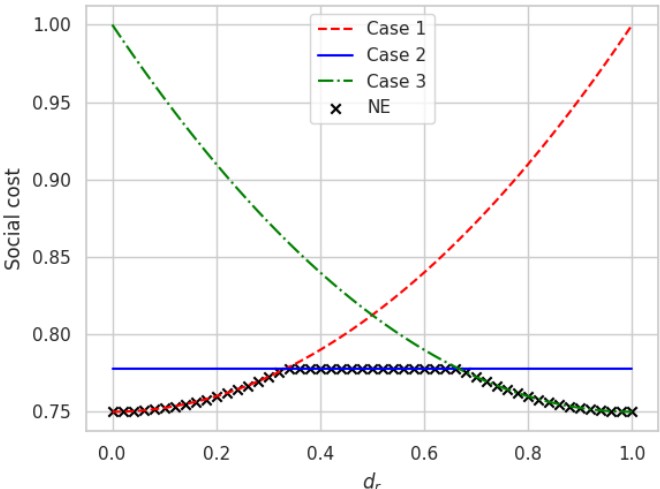

**Figure 3.** Social cost of Pigou routing with two route planners and $p = 1$.

As choices are independent, similar reasoning applies for more route planners. The optimal routing remains the same as the three cases in Example 1, but the effect of adding another selfish agent increases the worst possible cost.

**Example 2** (Three route planners). *Now suppose three route planners control the flow on the same Pigou network. As before, each route planner $r \in \{1, 2, 3\}$ performs a minimisation over their routing choice $x_r$. Since they chose their routing independently of one another, the same reasoning can be used to consider more populations. The optimal routing remains the same, but the effect of adding another selfish agent increases the worst possible cost. This can be seen in Figure 4, where the same behaviour of two operating systems is seen with another dimension.*

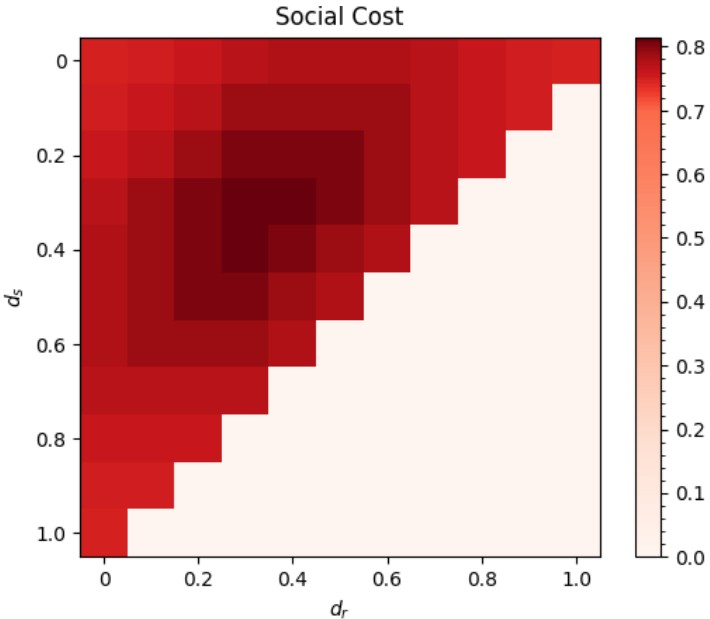

**Figure 4.** The social cost of routing on the Pigou network with $p = 1$ for three route planners.

Let us formalise the result that proportionality is linked to a high PoA. Once again, since the PoA is independent of network topology, we can find the worst-case example of it using the Pigou example. Thus, Examples 1 and 2 found the worst-case ratio of selfish route planners to fully cooperative route planners for two and three route planners, respectively. Let the number of route planners be $|R| \in \mathbb{N}$. We can find the PoA using the same method as the example for general $|R|$.

**Proposition 3.** *The PoA of a network control game is highest when the game is proportional.*

**Proof.** To find the worst-case PoA of route control, we want that no route planner is acting socially optimally. We can find the worst-case of routing on the Pigou example since it is independent of topology. Thus, we solve the minimisation problem

$$\min_{0 \le x_r \le d_r} \left[ x_r (\sum_{s \in R} x_s)^p + (d^r - x_r) \right]$$

To do so, we use the following Lagrangian function

$$L(x_r, \lambda_1, \lambda_2) = x_r (\sum_{s \in R} x_s)^p + d^r - x_r - \lambda(d^r - x_r)$$

The corresponding Karush–Kuhn–Tucker conditions are:

$$\frac{\delta L}{\delta x_r} = (\sum_{s \in R} x_s)^p + p x_r (\sum_{s \in R} x_s)^{p-1} - 1 + \lambda = 0$$

$$\lambda \frac{\delta L}{\delta \lambda} = \lambda(x_r - d^r) = 0$$

For general $p \ge 0$ and $|R| > 2$, the three cases remain the same as Example 1. The best response to $x_r = d^r$ is to choose $x_s = (1+p)^{\frac{-1}{p}}$, and when $x_r = x_s$, we have $x_r = (p|R|^{p-1} + |R|^p)^{\frac{-1}{p}}$. For no route planner to choose the socially optimal routing in Pigou's example, each route planner must have a proportion of control of population $i$ of at least $(p|R|^{p-1} + |R|^p)^{-1/p}$ and less than or equal to $1 - (p|R|^{p-1} + |R|^p)^{-1/p}$. For all $|R|$ and $p$, $(p|R|^{p-1} + |R|^p)^{-1/p} \ge \frac{1}{|R|}$. As $|R| \to \infty$, $(p|R|^{p-1} + |R|^p)^{-1/p} \to \frac{1}{|R|}$. Thus, the worst-case equilibrium cost can be achieved through a proportional assignment of populations. $\square$

The maximum social cost of Nash equilibria of the network control game also occurs for other distributions of route planner control. From Figure 3, we see that there is a set of population controls that maximise the social cost existing around the proportional version of the game. This set is characterised by each route planner having a share of control of at least $(|R|^p + p|R|^{p-1})^{-1/p}$ for each population. For example, with linear cost functions and two route planners, each route planner must control at least $1/3$ of each population, or for three route planners, they must control at least $1/4$.

We will now find the worst-case PoA for a network control game for polynomial edge-cost functions.

**Theorem 1.** *The worst-case PoA for a network control game with $|R|$ route planners and polynomial edge-cost functions at most degree $p$ is*

$$\frac{1 - |R| \left( |R|^{p-1}(p + |R|) \right)^{-\frac{1}{p}} + |R|^{p+1} \left( |R|^{p-1}(p + |R|) \right)^{-\frac{p+1}{p}}}{1 - (p+1)^{-\frac{1}{p}} + (p+1)^{-\frac{p+1}{p}}}$$

**Proof.** By Proposition 3, the worst-case equilibrium can be found when the game is proportional. Thus, we let each route planner solve the objective function

$$\min_{x_r} \left[ x_r \left( \sum_{s \in R} x_s \right)^p + d^r - x_r \right]$$

At the minimum, we have

$$\left( \sum_{s \in R} x_s \right)^p + x_r p \left( \sum_{s \in R} x_s \right)^{p-1} - 1 = 0.$$

Since the strategy spaces are symmetric and the game has an exact potential function, there exists a Nash equilibrium where each route planner plays the same strategy. The Nash equilibria of an exact potential game all have the same social cost, so this instance is also the worst Nash equilibrium. Thus,

$$(|R|x_r)^p + x_r p (|R|x_r)^{p-1} - 1 = 0.$$

which rearranges to

$$x_r = (p|R|^{p-1} + |R|^p)^{-1/p}.$$

The social cost of the worst-case Nash equilibrium is

$$|R|^{p+1}(p|R|^{p-1} + |R|^p)^{-1-1/p} + 1 - |R|(p|R|^{p-1} + |R|^p)^{-1/p}.$$

The social optimum of the game is where the total congestion on the bottom edge is $(p+1)^{-1/p}$, with a social cost of

$$(p+1)^{-1-1/p} + 1 - (p+1)^{-1/p}.$$

The ratio of these two costs gives us the result. □

For $|R| = 1$, by Theorem 1, the PoA is 1. Thus, the system is efficient when a route planner has full control of all vehicles. As $|R| \to \infty$, Theorem 1 implies that the PoA tends to that of the NIC congestion game it controls [18],

$$\frac{(p+1)^{\frac{1}{p}+1}}{(p+1)^{\frac{1}{p}+1} - p}.$$

Figures 5 and 6 plots the PoA as a function of $p$ for the network control games with varying $|R|$ and $p$.

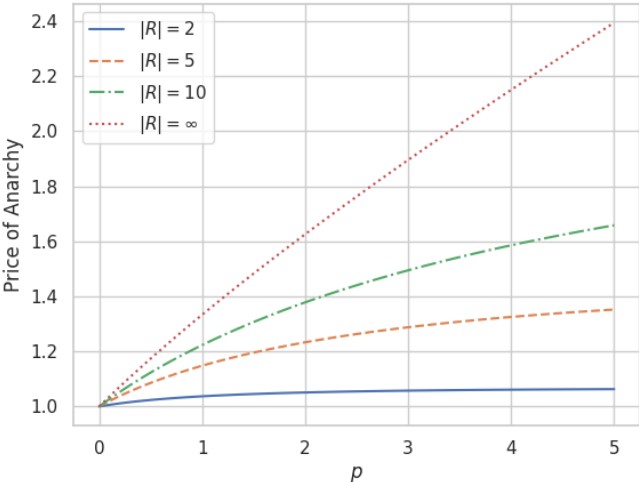

**Figure 5.** The PoA for a network control game for a range of $p$ and a selection of $|R|$ values.

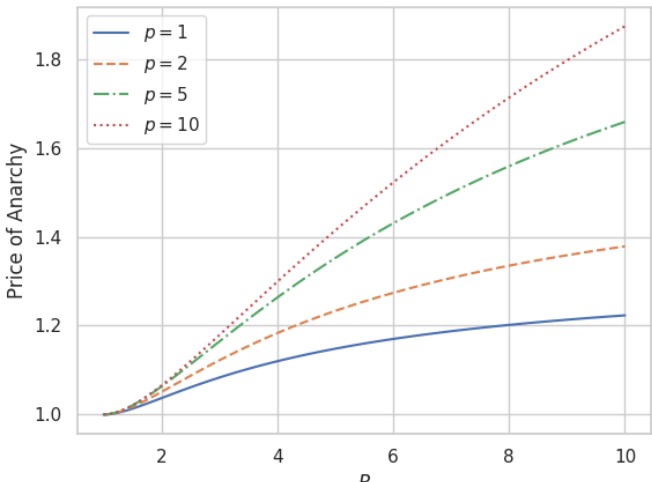

**Figure 6.** The PoA for a network control game for a range of $|R|$ and a selection of $p$ values.

The PoA for the network control game is significantly better than that of the NIC congestion game (where $|R| = \infty$) for a small number of route planners $\forall p > 0$. The system gets more inefficient as the number of route planners increases.

## 6. MARL Implementation

In this section, we consider the application of the theory of network control games to multi-agent reinforcement learning to test whether self-interested learning agents will converge to equilibrium strategies. To do so, we consider an instance of the Braess network with cost functions known to induce suboptimal selfish routing. The cost functions are shown in Figure 7.

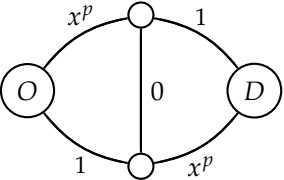

**Figure 7.** The Braess example where $d = 1$. With $p = 1$, the cost functions are linear, and for $p = 2$, the cost functions are quadratic.

To show that these results align with multi-agent learning, we simulated an instance of the network control game on this example for linear and quadratic edge-cost functions. We chose a proportional game, since this case has worst-case selfish-routing as indicated by Proposition 3.

The repeated game can be seen as an MDP by defining the following state, action, rewards, and transitions. The state is the congestion of the network, i.e., the flow on each edge. Note that the route planners have full information of the network congestion. The actions of the route planners are to select the demand of knowledge types. Since we restrict our analysis to the case where recommended knowledge types have only one route available, the action space is equivalent to the demand of routes. Figure 7 shows the four routes available to the population: along the two upper edges; along the lower two edges; and the two paths that include an upper edge, a lower edge, and the middle connecting edge. The route planners receive a reward equal to the negative of their cost function. Finally, the subsequent state is the user equilibrium of the congestion game. Thus, state change is deterministic from the actions of route planners and vehicles. Since there is no change in state that is external to the players in a repeated game, they are often referred to in MARL literature as stateless games. Note that we use a model-free algorithm which means we do not explicitly use the transition function in the learning algorithm.

We used the Asynchoronous Advantage Actor–Critic (A3C) algorithm [46], due to its use in multi-agent RL social dilemma environments, e.g., [48,49]. We simulated instances with either one, two, or three route planner agents controlling the flow. Each game consisted of playing the network control game shown in Figures 8 and 9 for 100 repeated rounds with no discounting ($\gamma = 1$). Thus, the SO cost is 150 or 123 for linear and quadratic costs, respectively. In both instances, the worst possible cost is 200. Each instance was averaged over three different random seeds. The neural network consisted of two fully connected layers of size 32 and a Long Short-Term Memory (LSTM) recurrent layer [50]. This network architecture was taken from [48]. We used the Ray library (https://github.com/ray-project/ray, accessed 10 January 2021) for a standard implementation of A3C with default parameter settings.

The learning curves for these experiments are shown in Figures 8 and 9. The results indicate that the agents learn to play strategies with a total cost that is close to the predicted PoA (from Theorem 1) for the edge-cost type and number of agents. Thus, reinforcement learning agents are vulnerable to choosing suboptimal routing as predicted by the theory. This suggests that the application of RL to route control requires cooperation, as with other social dilemmas, between route planners to minimise congestion.

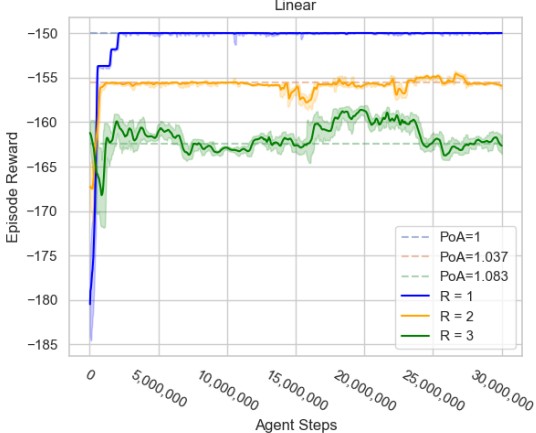

**Figure 8.** Learning curves for A3C agents playing a network control game on Braess' example for linear cost functions.

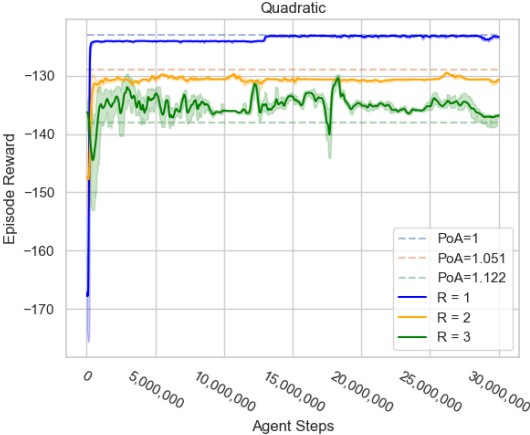

**Figure 9.** Learning curves for A3C agents playing a network control game on Braess' example for quadratic cost functions.

## 7. Choosing Route Planners

So far, we have studied vehicles that are assigned to route planners controlling their choices. Suppose that instead, we allow vehicles to strategically select their route planner prior to their journey. In this extension, Nash equilibrium outcomes are such that no vehicle

has an incentive to unilaterally deviate from the route planner they selected, given the prescribed route choices.

A *route planners game* can be a tuple $(\mathcal{M}, R)$, where $\mathcal{M}$ is an NIC congestion game, and $R$ is the set of route planners. Furthermore, the strategy space of players in $\mathcal{M}$ is $R$, since their routing is selected by the route planner they choose. Let $y_r^i$ indicate the share of control of $r \in R$ selected by population $i \in N$. Then, a strategy profile $\boldsymbol{y} = (\boldsymbol{y}^i)_{i \in N}$ is feasible if $\forall i \in N$, $\sum_{r \in R} y_r^i = d_i$. Each feasible $\boldsymbol{y}$ has a corresponding network control game where $\forall r \in R$ and $\forall i \in N$, $y_r^i = d_i^r$ and $i \in N_r$ if $y_i^r > 0$. Thus, each $\boldsymbol{y}$ has an essentially unique Nash equilibria $\boldsymbol{d}$ deciding the distribution of information. Define the cost function of a vehicle $i \in N$ to be

$$C_i(\boldsymbol{y}) := \sum_{r \in R} y_r^i \sum_{(i,k) \in \kappa_r} C_{(i,k)}(\boldsymbol{x}) d_{(i,k)}^r \mathbb{1}_{(i,k) \in K_i},$$

where $\boldsymbol{x}$ is the ICUE that results from $\boldsymbol{d}$. Moreover, a Nash equilibrium is $\boldsymbol{y}$ such that $\forall i \in N$ $C_i(\boldsymbol{y}) \le C_i(y', \boldsymbol{y}) \; \forall y' \in R$.

**Proposition 4.** *A route planners game is an exact potential game for potential $\Phi$ defined as*

$$\Phi(\boldsymbol{y}) := \sum_{e \in E} \int_0^{f_e(\boldsymbol{x})} c_e(z) dz,$$

*where $\boldsymbol{x}$ is the ICUE formed from $\boldsymbol{d}$ and $\boldsymbol{y}$.*

**Proof.** Consider the change in potential function between strategy distributions $\boldsymbol{y}$ and $\boldsymbol{y}' = (y_j', \boldsymbol{y}_{-j})$ for some $j \in N$, with respective ICUE profiles $\boldsymbol{x}'$ and $\boldsymbol{x}$.

$$\Phi(\boldsymbol{y}') - \Phi(\boldsymbol{y}) = \sum_{e \in E} \int_0^{f_e(\boldsymbol{x}')} c_e(z) dz - \sum_{e \in E} \int_0^{f_e(\boldsymbol{x})} c_e(z) dz$$

Rewrite as a sum over possible strategies in $S$,

$$= \sum_{i \in N} d_i \sum_{(i,k) \in K_i} \sum_{s \in S_{(i,k)}} \left[ x_s'^i \sum_{e \in s} \int_0^{f_e(\boldsymbol{x}')} c_e(z) dz \right.$$
$$\left. - x_s^i \sum_{e \in s} \int_0^{f_e(\boldsymbol{x})} c_e(z) dz \right]$$

Rewrite as a sum over the route planners' strategies,

$$= \sum_{i \in N} \sum_{r \in R} \sum_{(i,k) \in \kappa_r} d_{(i,k)}^r \mathbb{1}_{\{(i,k) \in K_i\}} \left[ y_r'^i \sum_{e \in K_i} \int_0^{f_e(\boldsymbol{x}')} c_e(z) dz \right.$$
$$\left. - y_r^i \sum_{e \in K_i} \int_0^{f_e(\boldsymbol{x})} c_e(z) dz \right]$$

Since the only difference between $y_r'^i$ and $y_r^i$ is when $i = j$,

$$= \sum_{r \in R} \sum_{(i,k) \in \kappa_r} d_{(i,k)}^r \mathbb{1}_{\{(i,k) \in K_j\}} \left[ y_r'^j \sum_{e \in K_j} \int_0^{f_e(\boldsymbol{x}')} c_e(z) dz \right.$$
$$\left. - y_r^j \sum_{e \in K_j} \int_0^{f_e(\boldsymbol{x})} c_e(z) dz \right]$$

$$= \sum_{r \in R} \sum_{(i,k) \in \kappa_r} d^r_{(i,k)} \mathbb{1}_{\{(i,k) \in K_j\}} \left[ y'^j_r C_{(i,k)}(\mathbf{x}') - y^j_r C_{(i,k)}(\mathbf{x}) \right]$$

$$= C_j(\mathbf{y}') - C_j(\mathbf{y})$$

Thus, $\Phi$ is an exact potential function. By definition, the network control game is an exact potential game. $\square$

Thus, Corollary 2 follows.

**Corollary 2.** *For every route planners game, there exists a Nash equilibrium, and this is essentially unique.*

Now, suppose we have an NIC congestion game with a socially inefficient UE and at least two route planners controlling the flow. Any route planner that has a small share of control of a population will choose the same strategy as players in a congestion game (selfish routing). Similarly, any route planner with a large share of control of a population plays by routing according to the social optimum. Since the UE of the game is socially inefficient, we know that the players choosing the route planner with a large share of control will have a strictly greater cost than those choosing a route planner with a small share of control. Thus, vehicles choosing their route planners have an incentive to choose the one with the least control. Any route planner that has less control over the population than any other route planner is more desirable to vehicles. Thus, there cannot be a route planner with strictly less control than all other route planners at the Nash equilibrium. We have ruled out the case where a route planner has no control over any population, so the flow must be proportional at the equilibrium.

**Proposition 5.** *Each Nash equilibrium of route planner games is proportional.*

**Proof.** Any route planner with control of a population less than $(p|R|^{p-1} + |R|^p)^{-1/p}$ will choose the same inefficient selfish routing as the vehicles of the NIC congestion game. Since this is the UE of the game, the other routing must be greater than or equal to this cost. Thus, vehicles prefer to choose a route planner with less than $(p|R|^{p-1} + |R|^p)^{-1/p}$ control over their population. Since $(p|R|^{p-1} + |R|^p)^{-1/p} \geq \frac{1}{|R|}$, the best-response dynamics will end when all route planners have proportional control of all populations. $\square$

Following from Proposition 3, we see that allowing vehicles to choose their route planner gives the worst possible PoA.

## 8. Conclusions

We studied multiple route planners optimising the routing of subpopulations in a nonatomic information-constrained congestion game through resource allocation. As their number grows, the equilibrium changes from achieving socially optimal routing to achieving the same inefficient routing as the original congestion game. We found the exact bound on the PoA of the induced game for polynomial edge-cost functions. Then, we used a simple example to show that MARL suffers from this PoA in practice. Additionally, we allowed vehicles to choose their route planner and showed that this only increases the overall inefficiency. Thus, companies using MARL routers to ease congestion require further incentives to cooperate with each other.

Natural extensions include analysing games with partial route planner control and the rest as selfish players with full or partial information which, we believe, could influence how autonomous vehicles design their route choice when the roads have a mix of human-driven and autonomous vehicles.

Another line of further work is to discover under what conditions is there an incentive to follow a route planner over autonomous routing. Designing incentive mechanisms for drivers to choose route planner control whilst achieving some level of fairness can impact

the use of route controllers in real-world traffic. Survey research into people's beliefs about the ethics of autonomous vehicles found that although people were in favour of a utilitarian approach of saving more lives over less, they also said that they would not purchase a utilitarian car themselves due to the risk of self-sacrifice [51]. This phenomenon was coined "the social dilemma of autonomous vehicles". Perhaps people would have a similar perspective of socially optimal routing—desiring a utilitarian system that is beneficial for everyone yet irrationally choosing the opposite. In which case, designing a routing system that drivers have no incentive to defect from, by choosing their own routes, would be an important extension of the work.

The results from this paper theoretically support the implementation of a centralised route planning algorithm to guide autonomous vehicles and reduce congestion. The higher the number of route planners controlling vehicles on the roads, the larger the inefficiency of the resulting routing equilibria. This is also the case for navigation apps; the more applications available to drivers, the worse the outcome of selfish routing will be. However, suboptimal routing could be mitigated if route planners cooperate with each other. The problem constitutes a social dilemma, so if route planners were able to detect if their rivals were cooperating or defecting, algorithms such as ARCTIC [49] could be adapted and utilised for safe cooperation in route control.

**Author Contributions:** Conceptualization, formal analysis, and original draft preparation was conducted by C.R. Supervision, review, and editing of writing was conducted by P.T. All authors have read and agreed to the published version of the manuscript.

**Funding:** This research received no external funding.

**Conflicts of Interest:** The authors declare no conflict of interest.

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
