# Peer review of "Fighting for Routes: Resource Allocation among Competing Planners in Transportation Networks"

_games, doi:10.3390/g14030037_

Round 1

Reviewer 1 Report

The paper is very saturated by different approaches and methods including algorithmic game theory, inefficiency of equilibria, potential and congestion games, resource allocation, transportation network games, and reinforcement learning. The considered setup is quite sophisticated. First, several social planners can control the information for separate agents (vehicles) so that to minimize social cost for the respective subpopulations of agents. Second, the agents can choose the social planners. Some interesting analytical results are received and supported by numerical calculations according to an Asynchronous Advantage Actor-Critic algorithm implementation. Namely, the authors studied multiple route planners optimizing the routing of subpopulations in a non-atomic information-constrained congestion game through resource allocation. As their number grows, the equilibrium changes from achieving socially optimal routing to achieving the same inefficient routing as in the original congestion game. The results can be applied to mitigate inefficiency issues arising in large transport networks with route controlled autonomous vehicles.

Remarks.

  1. It seems useful to give some examples (at least one) of practical interpretation of the competing social planners.
  2. I would recommend to include to the literature review the papers by Vladimir V. Mazalov from the Karelian Scientific Center RAS and by Victor V. Zakharov from the Saint Petersburg State University on network transportation games.
  3. In the first formula in Example 1 brackets are required after min (and elsewhere in the text).

Author Response

Reviewer one made three comments for suggestions.

The first comment was to include an additional example of practical interpretations which has been included at the beginning of Section 4.

The second comment was to include a paper reference but it was not clear which paper the reviewer thought was relevant and we could not find a relevant paper by the stated authors that should be added.

The final comment was a minor typo which has been corrected.

Reviewer 2 Report

In this paper, the authors have proposed multiple route planners to optimize the routing of subpopulations in a congestion game through resource allocation. The authors have found exact bounds of the induced game for polynomial edge-cost for Price of Anarchy, and shown how multi-agent reinforcement learning algorithms can suffer from finding an exact solution to the original problem of  choosing efficient route planner for vehicle. In general, the technical contributions of the paper are suitable for publications. The paper is also well organized with good presentation structure.

Some minor comments:

  • A notation table would be helpful.
  • On page 7 line 266, tuple definition has misalignment.
  • The paper can be re-organized in terms of putting some proofs into appendix rather than inside the text.
  • The related work section can be improved by comparison with the proposed contributions in this paper.

Author Response

Reviewer two suggested including a notation table and reorganising proofs into an appendix. We feel that the preliminaries and first few paragraphs of Section 4 should suffice for the reader to access relevant notation information. We chose not to include an appendix to make the flow of the paper easier. They also noted a misalignment of an equation which is fixed once the reviewing mode of the paper (paracol) is removed.

Reviewer 3 Report

I have been given the opportunity to review the paper titled "Fighting for routes: resource allocation among competing planners in transportation networks.”

The paper is very engaging, straightforward, and well-written. Building on the previous findings, the authors extend the theoretical framework for resource allocation in network control games. The authors prove that the worst-case PoA can be calculated and generalized for a network control game with R route planners and polynomial edge-cost functions at most degree p. Additionally, they show the relevance of the finding concerning PoA with varying Rs and ps, as well as the extension to multi-agent reinforcement learning.

Here are a few comments regarding possibilities for improvement that require minor revisions.

  1. The section Preliminaries could use additional sentences between the paragraph that would bond the paragraphs as building blocks in developing the idea and explain how the stated approaches combine into a big picture (i.e., why are they introduced).
  2. In line 170, the authors state, “is feasible if.” For such sentence formulation, the authors should prove if it is feasible or if it holds or not. However, the authors might want to rephrase it with “for,” e.g., the equation (1) holds for…
  3. Line 172 – please check the expression; it seems that s k before ∈K is missing.
  4. In lines 202-207, a reference to reinforced learning is missing.
  5. It seems that there should be a reference for lines 272-273.
  6. Check the consistency of x accentuation in lines 286-287.
  7. In line 290, the authors refer to Theorem 1, but Theorem 1 occurs in line 390. So, Corollary 1 cannot follow from Theorem 1, as Theorem 1 has not yet been introduced at this point. Please, check the line of thought and support it properly.
  8. The authors should state which software they used for simulations. In addition, the authors might want to add the code used (in a supplementary file) for the findings’ replicability.
  9. Given that the authors heavily focus on game efficiency, they might consider adding efficiency as a keyword.
  10. The authors conclude that the findings favor a centralized route planning algorithm. Perhaps a sentence or two could be added on the social aspect of the possibility for the application of such an approach and a possibility of its adoption by drivers (as it has been done in lines 491-502 for ethical beliefs), regarding other behavioral and social aspects, such as the perceived lack of options and personalization, different preferences, etc.

Author Response

Reviewer three asked for better flow in the preliminaries and we added a few sentences to aid this at the start of the section and when introducing reinforcement learning.

Comment 2 refers to the definition of feasibility which we chose not to change as this is standard terminology in the literature.

Comment 3 refers to a minor typo in notation, which we corrected.

From comment 4 we added a reference to further reinforcement learning resources.

Comment 5 refers to referencing the term "proportional" which is common in fairness literature but to our knowledge has not been used for congestion games in this way so we did not include a citation here.

We checked the notation in comment 6 for consistency.

Comment 7 refers to a reference to Theorem 1 from Acemoglu et al. so we changed the wording to make this clear.

Comment 8 refers to accessing source code, we choose not to include the reference in the paper but the code used to create these results is available on Github, should readers wish to find it.

We included "efficiency" as a keyword as suggested by comment 9.

Finally, we think that the reader's last comment is covered by the conclusion.